# Autophagy: Shedding Light on the Mechanisms and Multifaceted Roles in Cancers

**DOI:** 10.3390/biom15070915

**Published:** 2025-06-22

**Authors:** Hongmei You, Ling Wang, Hongwu Meng, Jun Li, Guoying Fang

**Affiliations:** 1Department of Pharmacy, Hangzhou Women’s Hospital, Hangzhou 310000, China; 2Department of Pharmacy, Shangyu People’s Hospital of Shaoxing, Shaoxing 312000, China; 17356516051@163.com; 3Department of Pharmacy, The First Affiliated Hospital of Anhui Medical University, Hefei 230022, China; 4Inflammation and Immune Mediated Diseases Laboratory of Anhui Province, Anhui Institute of Innovative Drugs, School of Pharmacy, Anhui Medical University, Hefei 230032, China

**Keywords:** autophagy 1, ATG 2, cancers 3, cell death 4

## Abstract

Autophagy, an evolutionarily conserved self-degradation catabolic mechanism, is crucial for recycling breakdown products and degrading intracellular components such as cytoplasmic organelles, macromolecules, and proteins in eukaryotes. The process, which can be selective or non-selective, involves the removal of specific ribosomes, protein aggregates, and organelles. Although the specific mechanisms governing various aspects of selective autophagy have not been fully understood, numerous studies have revealed that the dysregulation of autophagy-related genes significantly influences cellular homeostasis and contributes to a wide range of human diseases, particularly cancers, neurodegenerative disorders and inflammatory diseases. Notably, accumulating evidence highlights the complex, dual role of autophagy in cancer development. Thus, this review systematically summarizes the molecular mechanisms of autophagy and presents the latest research on its involvement in both pro- and anti-tumor progression. Furthermore, we discuss the role of autophagy in cancer development and summarize advancement in tumor therapies targeting autophagy.

## 1. Introduction

The dynamic equilibrium between cell death and proliferation plays a crucial role in various pathological and physiological processes across different organisms. Furthermore, the maintenance of organelle integrity and protein homeostasis is critical for regulating cell viability and cellular homeostasis. Autophagy, a process first proposed by Christian de Duve in 1963 [1,2], is categorized into three main types in mammalian cells based on the mechanism of cargo delivery to lysosomes: microautophagy, macroautophagy, and chaperone-mediated autophagy [3,4]. Macroautophagy, often referred to simply as autophagy [5], primarily involves the stages of initiation, nucleation, and elongation [6]. As an adaptable and responsive metabolic process, autophagy enables rapid cellular responses to diverse stimuli, including environmental influences, hormones, and metabolic challenges [7]. Growing evidence indicates that the activation of autophagy is essential in a range of diseases, including neurodegenerative and inflammatory diseases. The potential benefits of promoting autophagy have gained increasing attention, particularly in the context of eliminating protein aggregates implicated in neurodegenerative disorders. However, the role of autophagy in cancer is more complex, depending on factors such as tumor stage, biology, and the tumor microenvironment.

According to global statistics from the World Health Organization’s International Agency for Research on Cancer [8], lung cancer has become the most prevalent cancer worldwide, surpassing breast cancer in newly reported cases. Furthermore, other highly prevalent cancer types include esophageal, colorectal, prostate, stomach, and liver carcinomas. An increasing number of studies has indicated a significant correlation between autophagy and various human malignant tumors [9,10,11]. Autophagy appears to play a dual role in cancer occurrence and development [12,13,14,15]. On the one hand, by participating in protein and organelle quality control, autophagy contributes to maintaining genomic stability and preventing cellular injury, chronic tissue damage, and excessive inflammatory responses. Moreover, it prevents the accumulation of oncogenic P62 protein aggregates, thereby suppressing tumor initiation and progression. Consequently, autophagy serves as a critical tumor suppressor mechanism, particularly in the early stages of tumor development [16,17]. On the contrary, autophagy can preserve mitochondrial function, minimize DNA damage, and enhance cancer cell viability to resist the adverse tumor microenvironment such as hypoxia and nutrient deficiency, thereby facilitating oncogenesis or leading to chemotherapy drug resistance in the later stage of tumors [12,16,18]. In addition, the impact of autophagy on tumor formation is both autophagy gene-specific and tumor-specific. For instance, in breast cancer, the expression levels of autophagy-related genes are differentially associated with specific molecular subtypes [19]. Early research on the *Becn1* gene in mice revealed that the whole-body hemizygosity of *Becn1* resulted in tumor initiation in the liver, lymphatic tissues, and lungs, while no such effects were observed in other tissues [14,20,21]. In non-small cell lung cancer, autophagy activation not only contributes to chemoresistance against various agents, such as gefitinib and paclitaxel but also accelerates tumor progression [22,23]. Conversely, Wang et al. reported that autophagy activation can suppress pancreatic ductal adenocarcinoma formation via enhancing the P53 signaling pathway [24].

Targeting autophagy has become a promising strategy in oncology, autoimmune diseases, and others. Hydroxychloroquine (HCQ), a well-established autophagy inhibitor, disrupts autophagic flux by alkalizing the intracellular environment, which leads to lysosomal dysfunction and autophagy inhibition. Furthermore, HCQ can modulate innate immunity and inflammation, suppress T cell activation, restore the balance between regulatory T cells and helper T cells, and limit the production of pro-inflammatory cytokines. These multifaceted effects have established HCQ’s clinical significance. He et al. utilized chloroquine, a related compound, and enhanced tumor-killing efficacy when combined with mild photothermal therapy [25]. However, the off-target effects of traditional modulators and the context-dependent role of autophagy in disease necessitate the development of more precise spatiotemporal intervention methods. Examples include membrane protein degradation based on the phase separation of AUTAB [26] or the combination of autophagy inhibitors with immune checkpoint inhibitors [27].

In this review, we provide a comprehensive overview of recent advances in the study of autophagy signaling pathways, with a focus on various human tumors and different stages. We also comprehensively discussed the application of autophagy modulators and explored potential future directions, which may offer novel therapeutic approaches for cancer prevention and management in the clinic.

## 2. Major Types of Autophagy

Autophagy comprises several major types, primarily differentiated by their cargo delivery mechanisms to the lysosome. The three most extensively investigated forms are microautophagy, macroautophagy, and chaperone-mediated autophagy (CMA) [28,29,30] (Figure 1).

Microautophagy is a process by which the lysosomal membrane invaginates or deforms, facilitating the direct engulfment of cytoplasmic contents into the lysosome [31]. Non-selective microautophagy involves tubular membrane invaginations that directly sequester cytoplasm and its components into lysosomes. Selective microautophagy specifically targets certain organelles, including peroxisomes (referred to as micropexophagy), nonessential nuclear constituents (known as piecemeal microautophagy of the nucleus), and mitochondria (termed micromitophagy), for lysosomal degradation [32]. Previous research has implicated microautophagy in the pathogenesis of neurodegenerative disorders, such as Huntington’s disease [33] and Alzheimer’s disease [34], as well as lysosomal glycogen storage disorders, including Pompe disease [33]. Further investigation is warranted to elucidate the connection between autophagy and cancer.

Macro-autophagy transports cytoplasmic cargo to the lysosome within a double-membrane vesicle, termed an autophagosome. This autophagosome subsequently fuses with the lysosome, forming an autolysosome. The defining morphological feature of macroautophagy, distinguishing it from other vesicle-mediated trafficking pathways, is the de novo formation of autophagosomes, rather than membrane-budding pre-existing organelles. In essence, autophagosomes are assembled through expansion, rather than emerging from vesicles already consisting cargo [35]. Autophagosome formation and turnover are orchestrated by autophagy-related genes (ATGs), which are highly conserved across different species and typically categorized into distinct phases: induction, formation of the autophagosome, enlargement and elongation of the autophagosome membrane, and closure and fusion with the lysosome, culminating in the degradation of intraluminal contents [32,36]. Subsequent sections will provide a comprehensive elucidation of the molecular mechanisms and unique characteristics of macroautophagy. Macroautophagy can mediate engulfment and degradation through both selective and non-selective processes [29]. Currently, various forms of selective macroautophagy have been investigated, including the degradation of mitochondria, endoplasmic reticulum (ER), lysosomes, peroxisomes, ribosomes, and microorganisms, etc. [37,38]. Selective organelle autophagy is crucial for maintaining cellular homeostasis, ensuring organelle integrity and optimal numbers, even under fluctuating environmental conditions and diverse stressors.

Unlike microautophagy and macroautophagy, CMA is a protein-specific form of autophagy. Proteins containing KFERQ-like motifs are initially recognized by the heat-shock cognate protein HSPA8/Hsc70. Subsequently, they are targeted for lysosomal degradation after binding to LAMP2A (lysosomal-associated membrane protein 2A) and translocation through a channel formed by the oligomerization of this particular protein [39,40].

Macroautophagy, the most classical form of autophagy, is also the most complex. It was the first to be discovered, and its mechanism is the most well-defined, making it synonymous with autophagy in a narrowest sense.

## 3. Process of Autophagy

Under physiological conditions, the contribution of cellular autophagy to maintaining homeostasis is typically limited. Autophagy is markedly enhanced, as evidenced by a rapid increase in autophagosome formation, in response to the deprivation of nutrients and various stressors. Early discoveries of ATG genes stemmed from genetic studies in yeast, illuminating the intricate molecular signaling pathway governing autophagy [41] (Figure 2). The autophagy process generally comprises the following phases: initiation, nucleation, and elongation [6].

### 3.1. Initiation

Autophagy initiation involves the formation of an isolation membrane, known as a phagophore, which is proposed to originate from lipid bilayers derived from the endoplasmic reticulum (ER), the trans-Golgi network, endosomes, mitochondria, and the plasma membrane [42,43,44]. However, the precise source of the phagophore in mammalian cells remains a subject of ongoing debate. Canonical macroautophagy is a multi-step process characterized by the sequential and selective recruitment of ATG proteins [45]. In yeast macroautophagy, autophagosome formation is regulated by a complex comprising Atg1-Atg13-Atg17- Atg31-Atg29 kinase [46,47]. In mammalian cells, the autophagic cascade is initiated under the physiological control of MTORC1 (mechanistic target of rapamycin kinase complex 1), which promotes the dephosphorylation of ULK1/2 (unc-51-like autophagy-activating kinase 1/2) and ATG13 [48]. ULK1/2 is the mammalian homolog of yeast Atg1, while ATG13 is the homolog of yeast Atg13. Furthermore, ULK1 and ATG13 co-assemble within a complex with RB1CC1//FIP200 (RB1-inducible coiled-coil 1) and C12orf44/ATG101. RB1CC1//FIP200 may exhibit orthologous characteristics to yeast protein Atg17 [49]. Notably, C12orf44/ATG101 directly interacts with ATG13 and plays a crucial role in macroautophagy [50]. The nutrient status influences the association between the ULK1/2 complex and MTORC1. In nutrient-rich conditions, MTORC1 phosphorylates the ULK1/2 complex and associates with it, thereby inhibiting its activation and subsequent autophagosome formation. However, MTORC1 disengages under nutrient deprivation, contributing to dephosphorylation and autophagy induction [51]. The suppressive effect of MTORC1 is counteracted by AMP-activated protein kinase (AMPK), which phosphorylates BECN1 and ULK1 in response to declining ATP levels [52,53].

### 3.2. Nucleation

Another supramolecular complex is the class III phosphatidylinositol 3-kinase (PtdIns3K) complex, which consists of PIK3C3/VPS34 (phosphatidylinositol-3-kinase catalytic subunit type 3), BECN1, ATG14, PIK3R4/VPS15 (phosphoinositide-3-kinase regulatory subunit 4), and NRBF2 (nuclear receptor binding factor 2) [54]. ULK1 promotes the autophagy by enhancing the activity of phosphatidylinositol-3-kinase in the PtdIns3K complex. This complex can be transported to the prospective site of autophagosome formation and is essential for phagophore nucleation. The phosphorylation of phosphatidylinositol by PtdIns3K generates PtdIns3P, which is crucial for macroautophagy in both mammals and yeast [54]. Numerous regulatory factors have been identified to interact with the BECN1-PIK3C3/VPS34 complex, including SH3GLB1 (SH3 domain containing GRB2-like, endophilin B1), UVRAG (UV radiation resistance-associated), AMBRA1 (autophagy and Beclin 1 regulator 1), BCL2, and RUBCN (rubicon autophagy regulator). SH3GLB1 [55], UVRAG [56], and AMBRA1 [57] enhance the enzymatic activity of PIK3C3/VPS34, whereas BCL2 and RUBCN suppress it [40,58,59]. Further research is needed to fully elucidate the roles of these interactors in the endocytic and autophagic process.

### 3.3. Elongation

In yeast, PtdIns3P binds to multiple proteins. Among these, Atg18 and Atg21 are autophagy-related proteins located at the pre-autophagosomal structure (PAS) [60]. Mammalian cells express two Atg18 homologues, WIPI1 and WIPI2. These orthologs also participate in autophagy and associate with phagophores under amino acid starvation conditions through their interaction with PtdIns3P [60,61]. Another protein that binds PtdIns3P in mammalian cells is ZFYVE1/DFCP1 [42]. Both WIPI2 and ZFYVE1 are recruited to the omegasome. WIPI2 binds to and is activated upon interaction with PtdIns3P. Upon activation, WIPI2 recruits the ATG12-ATG5-ATG16L1 complex to sites of phagophore assembly, thereby enhancing the conjugation of LC3 (microtubule-associated protein light chain 3) with phosphatidylethanolamine through ATG3 and facilitating phagophore expansion and closure [62]. LC3-II is the lapidated form of LC3-I. The membrane material for phagophore formation and autophagic membrane expansion originates from various cellular compartments, including the Golgi apparatus, ER, recycling endosomes, mitochondria, and plasma membrane [63]. Furthermore, relevant lipid bilayers are contributed by ATG9-containing vesicles. Ultimately, the autophagosome expands until it reaches full formation and subsequent closure. Fusion of the lysosome with the outer membrane of the autophagosome results in autolysosome formation. Following lysosomal digestion, the enclosed cytoplasmic contents are degraded into macromolecules and amino acids. These components are then transferred from the lysosomal membrane to the cytochylema and ultimately recycled for anabolism.

## 4. Links Between Autophagy and Cancers

The dual role of autophagy in cancer has been extensively documented, highlighting its multifaceted impact on cancer development, with the capacity to both promote and inhibit malignant cell proliferation [64,65,66]. Autophagy is believed to contribute to tissue homeostasis by eliminating damaged, misfolded, or aberrant proteins, as well as clearing impaired mitochondria and other cellular components. The relationship between metabolic abnormalities and degenerative and inflammatory diseases resulting from autophagy deficiency remains an area of active investigation. Although the genetic inactivation of the core autophagy machinery in human cancers is infrequent, animal models have demonstrated that autophagy deficiency can contribute to tumorigenesis [67,68]. The capacity of autophagy to eliminate damaged mitochondria decreases the production of ROS and inhibits glycolysis, thereby contributing to tumor suppression. Conversely, autophagy can impede apoptosis and enhance survival within the nutrient-deprived tumor microenvironment, thus facilitating oncogenesis. These opposing impacts of autophagy on stress response have led to the hypothesis that autophagy may initially suppress tumor growth by maintaining cellular homeostasis but transitions to a pro-oncogenic function as tumors progress [69]. While the primary goal of chemotherapeutic agents is to induce apoptosis in cancer cells, complete success has been limited. This can be attributed to the fact that cancer cells activate autophagy in response to therapy-induced stress. Consequently, autophagy reduces cellular functions and hinders apoptosis while promoting tumorigenesis [70]. The initial indication of autophagy’s involvement in cancer came from the identification of Beclin 1 as a potential tumor suppressor in breast cancer [71]. Furthermore, the earliest evidence suggesting that autophagy could promote cancer development emerged from observations that autophagy is upregulated in hypoxic tumor regions and tumor-induced inflammation and supports tumor cell survival [72]. Cancer has been extensively studied to understand the functional role of autophagy. The following review will comprehensively discuss the current research status and potential applications of autophagy in various cancer types (Figure 3).

### 4.1. Lung Cancer

Lung cancer is a heterogeneous disease, characterized by variations in both its histological and molecular profiles. These variances influence not only classification but also significantly impact prognosis and therapeutic strategies. Based on molecular characteristics, lung cancer is broadly classified into two main subtypes: non-small cell lung cancer (NSCLC) and small-cell lung cancer (SCLC). Notably, NSCLC accounts for approximately 85% of all lung cancer diagnoses [73]. The global incidence and mortality rates of lung cancer are increasing, accounting for over 25% of all cancer-related deaths. Lung cancer has re-emerged as the most commonly diagnosed type of cancer globally [8]. In recent years, targeted therapies have improved survival rates for a specific patient subgroup. However, lung cancer still exhibits one of the lowest 5-year survival rates compared to other types of cancer [74]. Despite advances in therapeutic approaches, lung cancer remains a significant challenge due to the development of drug resistance, metastasis, and limited long-term survival. Emerging evidence suggests a strong correlation between autophagy and lung cancer cell mobility, invasion, stem cell differentiation, and immune evasion. On one hand, autophagy can promote lung cancer cell survival, thereby favoring tumor initiation and progression. On the other hand, autophagy can induce apoptosis or cellular demise in lung cancer cells through various pathways, thereby impeding tumor progression [64,75].

Guo and colleagues reported that autophagy supports *Ras*-driven lung cancer cells by providing metabolic resources, thereby meeting their energy requirements and sustaining nucleic acid synthesis, ultimately enhancing cell viability [76]. Another study demonstrated that the forced expression of iASPP induced autophagic flux. Furthermore, iASPP overexpression promoted autophagy and tumorigenesis, as evidenced by the increased conversion of LC3-I to LC3-II in a SCID/NOD mouse model [77]. Thus, increased iASPP expression may serve as a therapeutic target and indicate a poor prognosis in lung cancer. The autophagic dependence of *Ras*-driven non-small lung cell carcinoma cells has been confirmed using *Atg7*-deficient cells. In the initial phases of cancer, autophagy inhibition may promote tumorigenesis, but this inhibitory effect is reversed as cancer progresses. Early research demonstrated that the *Atg7* deficiency led to the accumulation of defective mitochondria, which, in turn, activated *P53*-dependent proliferation arrest and significantly inhibited tumor growth in *Kras^G12D^*-driven NSCLC [78]. In the absence of *P53*, *Atg7*-deficient tumors exhibited lipid accumulation, and their derived cell lines displayed impaired fatty acid oxidation and lipid utilization defects, confirming that autophagy supports the survival of *Ras*-driven tumors by maintaining mitochondrial function and lipid catabolism [78]. Rao et al. observed a comparable dual role of *Atg5*-dependent autophagy in lung cancer [79,80]. They found that impaired autophagy promotes adenosinergic signaling through the Hif1α pathway and increases tumor infiltration by Tregs, thereby modulating inflammation and immune surveillance, which can, respectively, stimulate and regulate tumorigenesis. In established cancers, damaged autophagy leads to reduced mitochondrial respiration, elevated oxidative stress, and an intensified DNA damage response. The genetic knockout of *P53* suppresses cancer progression in autophagy-deficient tumors [80]. What’s more, Wang et al. [81]. reported that TRIM59 exerts dual functions in NSCLC by regulating *BECN1*-dependent autophagy through a dual mechanism: first, it inhibits the NFκB pathway to downregulate *BECN1* transcription; secondly, it modulates the TRAF6-mediated K63-linked ubiquitination of BECN1, thereby influencing the assembly of the BECN1-PIK3C3 complex.

Surgical intervention and chemotherapy are conventional treatment options for primary lung carcinoma. Numerous studies indicate that autophagy inhibition may enhance the sensitivity of lung cancer cells to chemotherapy, thus proposing novel strategies for synergizing autophagy inhibitors with chemotherapy [82]. Research by Wu et al. demonstrated that gemcitabine, in combination with cisplatin for lung cancer treatment, induces autophagy in tumor cells and modifies their chemosensitivity, suggesting that gemcitabine-triggered autophagy provides a protective mechanism against apoptosis in human lung cancer cells [83]. Therefore, combining gemcitabine with autophagy inhibitors may enhance the therapeutic efficacy against lung cancer and offer a potentially effective treatment strategy. Another similar study also indicated that autophagy promotes cisplatin resistance in adenocarcinoma cells via activation of the AMPK/mTOR pathway [84]. Interestingly, Zhang et al. found that the knockdown of Caveolin-1 increased the therapeutic sensitivity of lung cancer to cisplatin-induced apoptosis by inhibiting Parkin-related mitochondrial autophagy and activating the ROCK1 pathway [85]. Autophagy induced by Camptothecin (CPT) was shown to provide a safeguard against programmed cell death in lung cancers. Notably, treatment with 3-methyladenine to inhibit autophagy resulted in increased CPT-induced cytotoxicity and enhanced DNA damage levels in lung cancer cell lines [86]. As mentioned above, autophagy plays a dual role in the progression of various cancers. An increasing body of evidence demonstrates that promoting autophagy can effectively impede cancer development and improve prognosis. Apatinib, a classical small-molecule anti-cancer and anti-angiogenic drug, was reported to exhibit promising inhibitory properties on NSCLC by accelerating apoptotic and autophagic cell death via suppressing P62 and Nrf2 expression [87]. Fan et al. found that Bruceine D treatment in NSCLS cells increased the LC3-II/LC3-I ratio, which was associated with reduced P62 expression and promoted autophagosome generation, significantly suppressing lung cancer cell proliferation [88]. Lung cancer cells incubated with gitogenin, a saponin extracted from *Tribulus longipetalus*, showed evident autophagosome accumulation, a crucial factor in gitogenin’s suppression of lung cancer. Mechanistically, gitogenin-induced autophagy primarily involves the activation of AMPK and inhibition of AKT signaling pathways [89].

### 4.2. Breast Cancer

Breast cancer, the most prevalent malignancy in women, displays diverse outcomes based on histopathological characteristics and molecular profiles [90,91]. In the murine model, the absence of the tumor suppressor *Palb2* in mammary glands impairs DNA damage repair and redox regulation, leading to tumor development. In this specific model, tumorigenesis associated with Palb2 deficiency was attenuated by the monoallelic deletion of *Becn1*, which impairs autophagy [92,93]. Notably, the absence of *Becn1* did not exhibit this carcinogenic effect in mice lacking *P53*, suggesting that the role of Beclin 1 in tumor formation is contingent upon specific circumstances [93]. The overexpression of human epidermal growth factor receptor 2 (HER2), an oncogenic receptor tyrosine kinase, is a common genetic alteration found in approximately 20% of breast cancer patients and is linked to a poor prognosis [94]. Studies by Vega-Rubín-de-Celis et al. demonstrated that HER2 inhibits autophagy by directly binding to and inactivating Beclin 1, thereby promoting the development of HER2-positive breast cancer. Conversely, enhancing basal autophagy through genetic manipulation or the use of Tat-Beclin 1 peptide to disrupt the HER2/Beclin 1 interaction and induce autophagy effectively inhibits HER2-driven tumor growth [95].

Several evolutionarily conserved ATGs have been implicated in breast cancer development. For instance, Atg7 has been shown to suppress proliferation and migration while promoting apoptosis in TNBC cell lines. Furthermore, Atg7 inhibits EMT by restraining aerobic glycolysis metabolism in TNBC cells [96]. Claude-Taupin et al. [97] investigated the impact of suppressing Atg9A expression using shRNA and CRISPR/Cas9 techniques in MDA-MB-436, a TNBC cell line. They observed increased *Atg9A* mRNA expression in vivo. Their finding demonstrated that Atg9A inhibition suppressed in vitro cancer characteristics, indicating the potential of Atg9A as a novel therapeutic target for TNBC. Moreover, autophagy inhibition through the knockdown of *LC3* and *Beclin1* significantly suppressed the proliferation and metastasis of TNBC cells and induced apoptosis by blocking multiple oncogenic signaling pathways, including cyclin D1, PARP1, and uPAR/integrin-β1/Src, indicating that targeting Beclin1-mediated autophagy is a potential therapeutic strategy for TNBC [98]. Additionally, the downregulation of Beclin1, ATG5, and ATG7 reduced proliferation across various TNBC cell lines [99]. Trastuzumab, a monoclonal antibody, is a targeted treatment for HER2-positive breast cancer. However, many patients develop resistance to trastuzumab after an initial positive response [100]. Sphingosine kinase 1, a proto-oncogene associated with resistance against breast cancer treatment, can be effectively targeted by FTY720 [101]. Interestingly, FTY720 hinders autophagy and induces in trastuzumab-resistant breast cancer cells [102]. Studies have shown that the combined administration of trastuzumab and FTY720 leads to a significant reduction in tumor growth compared to individual treatments, with outcomes similar to those achieved with autophagy inhibitors. This suggests that FTY720 suppresses autophagy to overcome treatment resistance while simultaneously triggering apoptosis [103,104]. Similarly, inhibiting Med19 enhances the effectiveness of doxorubicin in breast cancer treatment by downregulating HMGB1 and subsequently suppressing autophagy [105]. HMGB1 is positively associated with autophagy levels, NFκB/P65 activity, and doxorubicin resistance in breast cancer cells, with its role in promoting resistance attributed to its ability to trigger autophagy [106,107,108,109]. Combining anthracyclines with autophagy inhibitors shows promise in addressing chemoresistance in TNBC [110]. Zhang et al. found that endoplasmic reticulum stress in breast cancer cells upregulates METTL3/METTL14 through XBP1s, driving enhanced m6A modification to stabilize ER-phagy genes and promote paclitaxel resistance. Targeting this mechanism can enhance chemotherapy sensitivity [111]. In addition, Chipurupalli et al. demonstrated that hypoxia-induced ER stress mediates ER-phagy through the BiP-FAM134B complex to maintain cancer cell survival [112]. Self-assembled nanosheets deliver ER-targeted photosensitizers. These nanosheets induce ER damage via photodynamic therapy, concurrently activate ER-phagy for the degradation of damaged proteins, and remodel the immunosuppressive microenvironment [113]. In summary, the emerging role of macroautophagy and ER-phagy in breast cancer highlights its dual role in tumor progression. Future research should focus on incorporating the modulation of organelle-specific autophagy into multi-target therapeutic strategies to dismantle the adaptive mechanisms of treatment-resistant tumors. Incorporating the modulation of organelle-specific autophagy into multi-target therapeutic strategies will be crucial for dismantling the adaptive mechanisms of treatment-resistant tumors.

### 4.3. Esophageal Cancer

Esophageal cancer is characterized by its aggressive nature, with a five-year survival rate of under 20%. This malignancy is primarily classified into two histological subtypes, esophageal squamous cell carcinoma (ESCC) and esophageal adenocarcinoma (EAC). Research indicates that autophagy can both promote and suppress tumor growth, depending on various cellular and microenvironmental factors associated with esophageal cancer. In ESCC, ULK1, a crucial serine/threonine kinase, is frequently targeted by cellular factors that induce autophagy vesicles through the modulation of multiple upstream signaling pathways. Furthermore, AMPK, a well-established autophagy inducer, activates ULK1 via phosphorylation, alongside the phosphorylation of interacting proteins [114,115,116]. Wu et al. demonstrated the involvement of the AMPK/ULK1 pathway in autophagy initiation triggered by MACC1, a factor associated with tumor metastasis in colon cancer [117]. The suppression of MACC1 inhibited autophagy in ESCC cells, and the addition of 3-methyladenine reversed the malignant characteristics induced by MACC1 in ESCC cells. Moreover, *MACC1* knockdown led to the inactivation of the AMPK-ULK1 signaling pathway, and metformin, an AMPK activator, could reverse MACC1-induced autophagy in ESCC cells [117].

LC3, a protein undergoing post-translational cleavage and lipidation prior to its incorporation into autophagosomes, serves as the most characteristic marker of autophagy in esophageal carcinoma [118]. In ESCC, numerous independent studies have investigated the correlation between LC3 expression levels and patient survival, revealing diverse associations, including positive, negative, and no significant relationships [119,120,121]. Moreover, the correlation between patient outcomes in EAC and LC3 staining patterns varied across three distinct categories: crescent/ring-like formations, diffuse cytoplasmic, and globular structures. To be specific, poor overall survival was observed in patients with a high expression of crescent/ring-like or globular LC3 staining or a low expression of the other LC3 staining patterns [122]. Beyond LC3, other autophagy-related proteins, such as Beclin 1 and P62, have also been implicated in the progression of esophageal cancer. Chen et al. elucidated a collaborative interaction between NFE2L2 and P62 in enhancing ESCC invasiveness and inducing EMT [123]. Furthermore, the administration of autophagy inhibitors 3-MA partially reversed the anti-metastatic effects observed upon P62 and/or *NFE2L2* knockdown. M. Weh et al. reported that the incidence of BECN1 expression loss was 49.0% in EAC patients, whereas only 4.8% of controls exhibited this phenomenon [124]. A significant negative correlation was observed between reduced Beclin 1 expression and both the tumor stage and histologic grade, indicating a potential tumor-suppressive role of Beclin 1. The induction of autophagy by rapamycin resulted in simultaneous increases in overall Beclin 1 levels and its phosphorylation, exhibiting cell line-specific patterns and contributing to prolonged cellular viability [124]. Zhang et al. demonstrated that the overexpression of Beclin 1 significantly inhibited the proliferation of esophageal cancer Eca109 cells and the growth of transplanted tumors in nude mice by activating autophagy, accompanied with increased LC3-II and decreased P62, confirming the tumor-suppressive effect of the Beclin 1-dependent autophagy pathway [125]. Therefore, further comprehensive explorations are warranted to elucidate the precise role of autophagy and autophagy-related proteins in esophageal carcinoma.

### 4.4. Colorectal Cancer

Colorectal cancer (CRC), a complicated disease, involves dysregulated cellular pathways, including autophagy, which exhibits context-dependent roles in tumorigenesis and tumor progression [126]. In the early stages of tumorigenesis, autophagy suppresses tumor formation by maintaining genomic stability and preventing the propagation of DNA damage. However, as tumor development progresses, autophagy can facilitate the advancement of tumors [127]. Numerous genes, proteins, miRNAs, and lncRNAs have been reported to regulate CRC progression via modulating autophagy. For instance, a clinical investigation revealed an association between reduced Atg5 expression and the absence of lymphatic vessel invasion in CRC patients [128]. BECN1 levels are significantly elevated in colorectal and gastric tumors compared to normal stomach and colon mucosal cells, with rates of 95% and 83%, respectively, suggesting that the high BECN1 expression may contribute to the development of both CRC and gastric tumors [129]. Zhang et al. transfected a *Becn1*-expressing plasmid into HCT-116 and HCT-15 cells, two kinds of colon cancer cell lines, and injected the *Becn1* transfectants into nude mice [130]. Subsequently, they observed the phenotype changes and investigated the involved signaling pathways. The results demonstrated that cell viability, migration, and invasions were significantly inhibited, and autophagy was induced in Becn1-overexpressed cells. Moreover, the overexpression of Becn1 resulted in an increased expression of LC-3II and CDK 4 while decreasing cyclin E expression in the cancer cell lines. The repression of colon cancer cell proliferation in xenograft models was observed upon the induction of apoptosis and inhibition of proliferation by in vivo Beclin 1 expression [130], whereas BECN-1 plays a complex role in CRC. Researchers found that higher BECN1 expression was observed in patients with surgically resected stage II and III colon tumors who underwent 5-FU treatment, leading to poorer overall survival. This suggests a potential role of autophagy in the development of drug resistance [131]. Survival rates were significantly higher among patients with tumors lacking ATG10 expression compared to those with tumors expressing ATG10. Furthermore, the level of autophagy-related protein ATG10 correlated with tumor invasion and metastasis [132]. Sakitani et al. found that autophagy inhibition in colon cancer cells led to the activation of P53 and UPR, which resulted in increased apoptosis and exhibited antitumor effects [133]. Both in vivo and in vitro studies have shown that the use of autophagy inhibitors can significantly enhance the effectiveness of sinoporphyrin sodium-mediated photodynamic therapy in treating colorectal cancer cells [134].

An increasing number of studies have demonstrated the involvement of miRNAs or lncRNAs in CRC development and chemoresistance by regulating autophagy [135,136,137]. Chemotherapeutic drugs are commonly employed as an adjuvant treatment for CRC [138]. Oxaliplatin (OXA), a third-generation platinum compound, has shown remarkable efficacy in CRC management [139]. However, despite the swift decrease in tumor size following chemotherapy, cancer cells frequently develop resistance to OXA, leading to cancer recurrence and metastasis [140]. Similarly, the ineffectiveness of 5-FU in treating CRC is largely attributed to the development of drug resistance. Several studies suggest a notable association between the protective impact of autophagy on cells and chemoresistance. Zhang et al. demonstrated that *miR-22* suppresses autophagy and enhances apoptosis, thereby improving the sensitivity and responsiveness of CRC cells to 5-FU treatment by silencing B-cell translocation gene 1 [141]. mTOR plays a crucial role in regulating the balance between apoptosis and autophagy, and its expression is controlled by *miR-338-3p*. The regulation of *miR-338-3p*-mTOR autophagy has been shown to be dependent on P53 and to play a role in the cellular response to 5-FU treatment [142,143]. Furthermore, one study indicated that *miR-125b* contributes to the development of 5-FU resistance by promoting autophagy, as evidenced by the upregulation of Becn1, the cleavage of microtubule-associated protein LC3-II, and the formation of autophagosomes [144]. Numerous other miRNAs, including *miR-409-3p* [145], *miR-183-5p* [146], *miR-27a* [147], *miR-20a* [148], and *miR-124* [149], etc., have also been implicated in CRC progression and drug resistance by regulating autophagy.

Compelling evidence underscores the critical role of lncRNAs in modulating oncogenic signaling pathways within CRC, particularly through their influence on autophagy. Linc-POU3F3 depletion in SW480 and LOVO CRC cells promotes autophagosome formation, potentially leading to the accumulation of LC3II, ATG5, ATG7, and BECN1. This effect may be partially attributed to the upregulation of SMAD4 levels. Clinically, *linc-POU3F3* expression correlates with adverse outcomes in CRC, such as an increased risk of recurrence and reduced recurrence-free survival, indicating that *linc-POU3F3*-mediated autophagy inhibition may exacerbate the malignant characteristics of CRC [150]. *LINC00858*, a cancer-promoting lncRNA, is highly abundant in both CRC cells and tissues. Silencing *LINC00858* induces autophagy, apoptosis, and senescence in vitro and inhibits tumor development in vivo by preventing methylation of the *WNK2* promoter and increasing WNK2 expression [151]. Furthermore, the inhibition of hypoxia-induced autophagy by *LncRNA CPS1-IT1* has been linked to the inactivation of HIF-1α, resulting in the suppression of EMT and CRC metastasis [152]. The expression of long non-coding RNA HOX transcript antisense RNA (*HOTAIR*) is significantly elevated in CRC patients and tumor cells following exposure to irradiation [153]. *HOTAIR* promotes cellular survival and autophagy while attenuating cellular apoptosis and radiation sensitivity by modulating the *miR-93*/ATG12 pathway in CRC. The expression of lncRNA TUG1 is also elevated in CRC cells and tissues. The increased level of TUG1, stabilized by IGF2BP2, promotes the proliferation, migration, and autophagy of CRC cells through the *miR-195-5p*/HDGF/DDX5/β-catenin pathway, consequently enhancing the resistance of CRC cells to DDP treatment in LS513 and LOVO cell lines [154].

### 4.5. Prostate Cancer

Prostate cancer is a leading cause of mortality among men in the United States. In 2021, an estimated 31,000 deaths were attributed to prostate cancer out of a total of 248,000 diagnosed cases [155,156]. Androgen-deprivation therapy is often an effective treatment option for prostate cancer [157,158]. However, the clinical management of prostate cancer is complicated by the development of castration-resistant prostate cancer [159]. Despite the fact that the pathophysiology of prostate cancer remains incompletely understood, numerous studies have implicated a dual role of autophagy in this disease [160,161]. Autophagy typically functions to maintain genome stability, eliminating carcinogenic substances, and thus promotes cellular homeostasis and safeguards cell DNA against harm. Thus, a significant proportion (40%) of prostate cancer patients exhibit mutations in the Beclin 1 gene leading to the elimination of specific proteins [162]. Conversely, under certain stress conditions, autophagy can be upregulated in prostate cancer cells, potentially promoting cancer cell survival. Reduced androgen levels have been shown to enhance autophagy in LNCaP cells, a human prostate cancer cell line sensitive to androgens. This phenomenon may contribute to prostate cancer progression and an androgen-independent phenotype, a common observation in clinical settings [163].

Given the strong association between autophagy dysregulation and prostate cancer, current research focuses on evaluating autophagy modulators for tumor treatment [164,165]. Rapamycin, also known as sirolimus, effectively enhances autophagy by suppressing mTOR1/2 protein [166]. Preclinical studies have shown that rapamycin administration improves prostatic inflammation and benign prostatic hyperplasia through stimulating autophagy. However, the efficacy of rapamycin and other mTOR inhibitors in prostate cancer remains a subject of debate. While rapamycin demonstrated significant inhibitory effects on the proliferation of both androgen-dependent and -independent prostate cancer cell lines by promoting autophagy, clinical trials evaluating rapamycin as a standalone treatment in prostate cancer patients have not shown significant reduction in cancer proliferation or progression [167,168]. Miyazawa et al. demonstrated that administration of simvastatin significantly elevated autophagy and inhibited cell proliferation in a dose-dependent manner in prostate cancer cell lines. The co-administration of simvastatin and rapamycin effectively triggered autophagy and further augmented the suppressive effect of simvastatin on cellular proliferation [169]. Everolimus, a synthetic rapamycin analogue, selectively suppresses mTORC1. Although preclinical studies have shown promising results in inhibiting prostate cancer progression in vivo and in vitro, clinical trials have indicated the limited effectiveness of everolimus monotherapy in reducing disease progression in prostate cancer patients [170,171]. One potential mechanism contributing to the limited effectiveness of mTOR inhibitors is the suppression of androgen receptors via the PI3K/AKT/mTOR pathway. Inhibition of this pathway can restore androgen receptor signaling, thereby promoting tumor progression [172]. Recent studies have provided evidence of the antiproliferative properties of metformin in prostate cancer cells. By targeting AMPK, metformin enhances LC3 expression and reduces P62 protein levels, indicating its potential role in activating autophagy in prostate cancer cells. In vivo studies in mice have also demonstrated the effectiveness of metformin in inhibiting tumor progression through autophagy activation [173,174]. Furthermore, the administration of valproic acid for 10 to 14 days has been observed to induce autophagy, activating Caspase-2 and Caspase-3, and subsequently leading to a decrease in prostate cancer proliferation. Prolonged valproic acid administration resulted in a notable reduction in the proliferation of tumor xenografts compared to the untreated control group [175]. Lin et al. found that the benzyl isothiocyanate treatment of prostate cancer cells decreased mTOR activity, induced autophagy, and enhanced cancer cell apoptosis [176].

### 4.6. Hematologic Malignancies

Similar to solid tumors, autophagy exhibits a dual role in hematological malignancies. On one hand, autophagy may promote drug resistance by supporting cancer cell survival through self-degradation. Conversely, autophagy can function as a tumor-suppressing pathway, facilitating robust antitumor immune responses and potentially protecting normal tissues from the adverse effects of cancer therapies. Numerous studies suggest that suppressing autophagy can significantly enhance the efficacy of anticancer agents. Schätzl et al. [177] demonstrated that Tyrosine kinase inhibitors (TKIs), such as nilotinib and imatinib, the gold standard treatment of chronic myeloid leukemia (CML), can induce autophagy in CML cells by inhibiting the PI3K-Akt-mTORC1 signaling pathway. However, this induced autophagy enhances the survival of CML stem cells, leukemia formation capacity, and resistance to TKIs [178,179]. Consequently, inhibiting autophagy increased imatinib-induced cell death in both cell lines and primary CML cells [180].

Emerging studies have further elucidated the role of autophagy in hematological malignancies. Ren et al. discovered that all-trans retinoic acid induces the differentiation of leukemia cells into normal granulocytes by activating autophagy, which targets and degrades the oncogenic fusion protein PML-RARα in acute promyelocytic leukemia [181]. Furthermore, researchers are investigating the potential of autophagy activation to mitigate the toxic side effects of chemotherapy. Le and colleagues suggest that rapamycin-induced autophagy may offer neuroprotection against proteasome inhibition [182], potentially addressing the neurotoxicity associated with proteasome inhibitors in multiple myeloma treatment.

### 4.7. Other Cancers

Beyond the cancers previously discussed, emerging evidence highlights the significant role of autophagy in gastric cancer [183], liver cancer [184], thyroid [185], cervical [186,187], bladder cancers [188], and various other tumors [14,189,190].

Autophagy can either promote or inhibit the progression of gastric cancer. Notably, autophagy can contribute to cancer progression originating from precancerous lesions in gastric tissue. Mommersteeg et al. demonstrated that the G allele of *ATG16L1 rs2241880* triggers protective autophagy during the development of gastric cancer from premalignant lesions [191]. Thus, suppressing autophagy in premalignant lesions may hinder gastric cancer development, whereas excessive autophagy activation in later stages can induce cell death in GC. Furthermore, IFN-γ elevates PD-L1 levels in gastric cancer, thereby enhancing carcinogenesis. The increase in P62/SQSTM1 levels and the upregulation of NF-κB contribute to immune evasion and the increased infiltration of tumor lymphocytes, which can be attributed to the inhibition of autophagy [192]. Recent findings suggest a dual role for autophagy in liver cancer: impeding initial formation while simultaneously promoting the progression and malignancy of pre-existing liver tumors. The initial confirmation of autophagy’s tumor suppressor role came from the seminal research by Beth Levine et al. They observed spontaneous tumor development in various tissues, including the liver, in aged mice with Beclin 1 heterozygosity [20,71]. A clinical study indicated that the reduced expression of BECN1 is associated with hepatocellular carcinoma (HCC) grade, suggesting its potential as a prognostic biomarker for HCC [193]. Further investigations revealed that *Atg5* or *Atg7* gene deletion leads to abnormal accumulation of the selective autophagy substrate P62, which competes with the Nrf2 binding site of Keap1 protein and blocks the ubiquitination degradation pathway mediated by Keap1, resulting in the persistent activation of the transcription factor Nrf2. Activated Nrf2 drives the excessive transcription of downstream target genes, thereby triggering a chronic inflammatory response and a cascade of liver fibrosis, ultimately promoting the occurrence and development of HCC [194,195,196]. These findings strongly support the role of autophagy as a genuine suppressor of tumor formation. Interestingly, mice with liver-specific knockout of *Atg5* or *Atg7*, which have impaired autophagy in their livers, exhibit the growth of non-cancerous adenomas rather than malignant HCC. Remarkably, mice with a specific knockout of *Atg5* in the liver do not exhibit HCC formation even following exposure to diethylnitrosamine, a known hepatic carcinogen [197], indicating that autophagy could potentially play a crucial role in facilitating the transition of benign tumors into malignant tumors during the advanced tumor progression. Moreover, suppressing autophagy can enhance the sensitivity of liver cancer cells to chemotherapy, suggesting that autophagy serves as a protective mechanism against cell death induced by chemotherapeutic agents [198]. Qu et al. reported that the overexpression of c-Myc activates mitophagy by inducing mitochondrial fission and P62 aggregation to remove damaged mitochondria and inhibit apoptosis, thereby leading to the resistance of hepatocellular carcinoma cells to cabozantinib. The inhibition of autophagy or P62 aggregation can reverse this drug resistance [199].

Papillary thyroid carcinoma (PTC) is the most prevalent form of thyroid cancer. Qin et al. observed reduced RBM47 expression in both PTC cells and tissues. Furthermore, RBM47 overexpression activated autophagy and inhibited proliferation in PTC cells [200]. Mechanistically, RBM47 upregulation hinders PTC cell proliferation and activates autophagy via binding with SNHG5. SNHG5 recruits USP21, which interacts with FOXO3 to impede its ubiquitination and enhance its nuclear translocation. By accelerating the transcription of *Atg3* and *Atg5*, *FOXO3* promotes autophagy and stimulates RBM47 transcription, thus establishing a positive feedback loop [200]. Numerous studies have confirmed the close association between cervical carcinogenesis and cellular autophagy, as well as autophagy-related proteins [201,202,203]. Moreover, autophagic activity can be influenced by HPV infection. HPV binds to host cells and engages with EGFRs located on the plasma membrane, leading to Akt phosphorylation, Pten inactivation, and the subsequent activation of the downstream mTOR signaling pathway. Consequently, cellular autophagy is inhibited, preventing early virus eradication and enhancing susceptibility to HPV acquisition [204]. Additionally, HPV infection can impact autophagy-associated DNA damage repairment [205]. Therefore, several agents have been developed to treat cervical cancer via targeting autophagy. For instance, Kayacan et al. demonstrated that the combined administration of apigenin and curcumin effectively suppressed cell proliferation in cervical cells by inducing the expression of autophagy-related genes such as *ATG5*, *ATG12*, *BCL-xL*, and *BECN1*, ultimately promoting autophagy-mediated cellular demise [206]. Luo et al. found that quercetin nanoparticles suppressed the proliferation, invasion, and migration of cervical cancer cells, similar to the function of JAK2 inhibition. In this process, cervical cancer cell death was facilitated via the induction of autophagy and apoptosis [207]. Notably, the involvement of autophagy in bladder cancer is becoming increasingly evident. Recent research indicates that autophagy plays a dual role in controlling bladder cancer progression. On one hand, autophagy promotes bladder cancer progression by supporting tumor cell survival and reducing their susceptibility to treatment. On the other hand, autophagy with pro-death properties intensifies apoptosis and other mechanisms leading to cell death in bladder cancer, thereby impeding tumorigenesis [208]. Dong et al. discovered that bladder cancer progression can be facilitated by increasing the levels of HK2, MMP-9, SLC2A1, MCT1, and MCT4 through the induction of autophagy in cancer-associated fibroblasts (CAFs). Conversely, hindering bladder cancer progression can be achieved by suppressing autophagy in CAFs via *ATG5*-siRNA to deactivate it [209]. Recently, advancements have been made in autophagy inhibition through chemical means, including the utilization of RNAi factors, pharmaceutical inhibitors, and compounds derived from natural sources with bioactive properties, such as chloroquine and 3-methyladenine [210,211,212]. The advantage of creating these autophagy regulators lies in the potential to amplify the cytotoxic effects of anti-tumor compounds and increase the sensitivity of cancer cells to chemotherapeutic drugs by inhibiting autophagy, particularly when it serves a protective role [213,214]. Du et al. revealed that 4-Methoxydalbergion, a compound isolated from *Dalbergia sissoo Roxb*, can repress the proliferation of bladder cancer cells by activating autophagy, accompanied by the increased expression of LC3-II/LC3-I and Beclin 1 [215]. To gain a deeper understanding of autophagy’s function, future investigations should thoroughly examine autophagy induction and suppression across different phases of bladder cancer.

## 5. The Possibility of Developing Autophagy Modulators for Cancer Treatment

As previously described, autophagy exhibits a complex and multifaceted role in cancer biology. Numerous studies have demonstrated that autophagy can promote cancer cell survival and proliferation, thereby contributing to tumor progression and therapeutic resistance [216,217]. For instance, in a soft microenvironment, autophagy has been shown to increase tamoxifen resistance in breast cancer cells [216]. Conversely, autophagy may also suppress tumor growth by inducing apoptosis in cancer cells [218,219]. Consequently, the development of drugs capable of precisely modulating autophagy represents a critical strategy for improving cancer therapy.

Classic autophagy modulators include 3-methyladenine (3MA), a well-established autophagy inhibitor. In studies designed to enhance the efficacy of umbilical cord blood natural killer (NK) cells against triple-negative breast cancer, 3MA effectively suppressed autophagy [218]. Furthermore, chloroquine and hydroxychloroquine are the only autophagy inhibitors currently available for clinical use. The survival period of 18 glioma patients who received combined treatment with chloroquine and anti-tumor drugs was significantly longer than that of the control group (33 months versus 11 months) [220]. Additionally, the preoperative combined use of hydroxychloroquine and anti-tumor drugs can reduce CA199 levels in pancreatic cancer patients by 61% [221]. Xuan et al. found that hydroxychloroquine enhances the anti-cancer activity of bevacizumab on glioblastoma by suppressing autophagy [222]. In patients with BRAF V600E mutant brain tumors resistant to vemurafenib (a BRAF inhibitor), the combination of the autophagy inhibitor chloroquine significantly reversed drug resistance, achieving tumor growth inhibition and increased cell death [223]. However, the clinical application of traditional autophagy modulators is often limited due to their non-specific accumulation and systemic toxicity. Cheung et al. demonstrated that the novel autophagy modulator YM155, in combination with BIRC5 inhibition, induces DNA damage by promoting autophagy-dependent ROS production and concurrently downregulates key homologous recombination repair proteins [224].

The antidepressant sertraline was shown to induce autophagy by targeting VDC1, activating AMPK, and inhibiting the mTOR signaling pathway, presenting a novel therapeutic approach for autophagy-related diseases. Cheng et al. developed GPX4-AUTAC, a system that precisely degrades the GPX4 protein through the autophagy pathway. By mimicking the natural ubiquitination–autophagy receptor recognition mechanism (TRAF6/P62), GPX4-AUTAC selectively induces the autophagy-dependent degradation of GPX4, efficiently triggering ferroptosis and inhibiting breast cancer growth [225]. Loperamide was demonstrated to specifically induce RE-TREG1/Tex264-dependent reticulophagy via the activation of ATF4-mediated endoplasmic reticulum stress, subsequently triggering autophagic cell death in glioblastoma [226]. The targeted protein degradation platform ATNC (phagosome), based on the autophagy pathway, utilizes nanobodies for selective target recognition. Through modular design, ATNC precisely degrades traditionally undruggable targets (such as HE4 protein), effectively inhibiting the proliferation and migration of ovarian cancer cells [227]. Given the complex role of autophagy in cancer and the limitations in the specificity and bioavailability of current autophagy regulators, this field continues to face significant challenges. Future research should prioritize the development of more specific and effective autophagy regulators while comprehensively investigating the molecular mechanisms underlying autophagy regulation, thus providing stronger guidance for clinical applications.

## 6. Discussion and Future Directions

As previously mentioned, autophagy actively participates in all stages of tumorigenesis, including tumor initiation, progression, development, as well as the maintenance of the malignant state. The function of autophagy in cancer is highly context-dependent. In the early stages of tumorigenesis, autophagy can function as a tumor-suppressive mechanism by removing damaged organelles and proteins [16,17,32,228]. Conversely, in established tumors, autophagy can promote cancer cell survival, proliferation, metastasis, and resistance to therapy by providing nutrients and enabling recycling under stress conditions such as hypoxia, nutrient deficiency, and treatment-induced damage [229,230,231,232]. Given its role in promoting survival in advanced cancers and contributing to therapy resistance, the primary therapeutic approach currently under extensive study and development involves blocking autophagy. This is often achieved through compounds like hydroxychloroquine or its derivatives, with the aim of enhancing tumor sensitivity to standard treatments or addressing resistance [233,234,235,236]. Human cancer cells implanted in immunodeficient hosts exhibited increased sensitivity to radiotherapy or chemotherapy when exposed to autophagy inhibitors such as 3-MA, HCQ, CQ, and wortmannin [237]. However, the optimization of combination strategies, administration sequences, and dosage regimens lacks a robust theoretical foundation and predictive models. Additionally, suppressing autophagy may result in adverse short-term or long-term consequences for cancer patients due to two primary factors. First, autophagy plays a crucial role in the survival, growth, and functional effectiveness of specific immune cell types associated with tumor regulation [238,239,240]. Second, theoretically, autophagy inhibition could increase the likelihood of healthy tissues undergoing carcinogenic changes or suffering other toxic effects. However, the specificity of many pharmaceutical autophagy modulators is limited. For example, wortmannin and 3-MA are both non-selective inhibitors of PI3K and can inhibit the catalytic activity of multiple PI3Ks, including those beyond VPS34 [241]. Therefore, exploring highly targeted regulators of autophagy is of great significance for developing clinically feasible strategies to regulate autophagy. Targeting specific complexes (such as ULK and VPS34), utilizing PROTACs/gene editing [242], and designing tumor-targeted delivery systems represent key approaches to advance the development of precise autophagy regulators. The intricate dual nature of autophagy presents substantial challenges and potential avenues for therapeutic intervention. Consequently, the therapeutic modulation of autophagy demands highly precise targeting based on tumor type, stage, genetic background, and concurrent treatments. In the foreseeable future, integrating conventional cancer treatment with autophagy modulators holds promise. Another challenge is that commonly used biomarkers for monitoring autophagy are not always applicable for monitoring autophagic flux (the degradation products of autophagosomes and their contents in lysosomes). For instance, the expression of lipidated LC3 or the amount of GFP–LC3+ puncta, which are used to statistically assess the size of the autophagosomal compartment, has been questioned because they only monitor static markers [243]. As an alternative approach, dual fluorescence-labeled LC3 has been effectively utilized to monitor intracellular autophagic flux in real time [240,241,244]. This technology is of vital importance for the clinical monitoring of the effects of autophagy regulators. 

## 7. Conclusions

The current research is centered on elucidating the dual mechanism of autophagy in cancer and developing selective targeting strategies. Looking ahead, breakthroughs in clinical application will be achieved by addressing its complexity through precise interventions and personalized treatment approaches. Although autophagy plays a dual role in most cancers, its beneficial effects are wildly acknowledged in various other medical conditions, such as neurodegenerative diseases (including Parkinson’s [230,245], Alzheimer’s [246,247,248], and Huntington’s diseases [249]) and ischemic heart disease, due to its ability to eliminate harmful substances and enhance cellular survival. Hence, autophagy has emerged as a novel and influential regulator of disease progression, attracting significant scientific interest and holding substantial clinical importance.

## Figures and Tables

**Figure 1 biomolecules-15-00915-f001:**
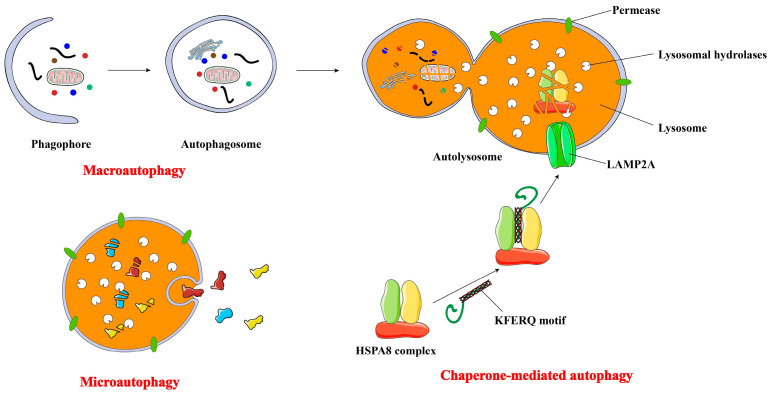
Major types of autophagy. During the process of microautophagy, the lysosomal membrane invaginates to envelop and degrade substrates within phagosomes, which are subsequently degraded within lysosomes. Different from microautophagy, macroautophagy entails the involvement of autophagosomes in the degradation process. During macroautophagy, damaged organelles and soluble macromolecules within the cytoplasm are enclosed by membranes originating from either the mitochondria or endoplasmic reticulum, leading to the formation of autophagosomes that are bounded by one or two membranes. The outer membrane of the autophagosome then merges with the lysosomal membrane, leading to the formation of an autolysosome. Within this structure, the materials designated for degradation through autophagy are broken down by various hydrolases. In the process of CMA, chaperone proteins identify and attach to soluble substrate proteins containing a particular amino acid sequence and then facilitate their transport to the lysosome. This process involves interacting with the LAMP2A receptor located on the lysosomal membrane. Once the substrate proteins reach the lysosome, they are broken down by lysosomal hydrolases.

**Figure 2 biomolecules-15-00915-f002:**
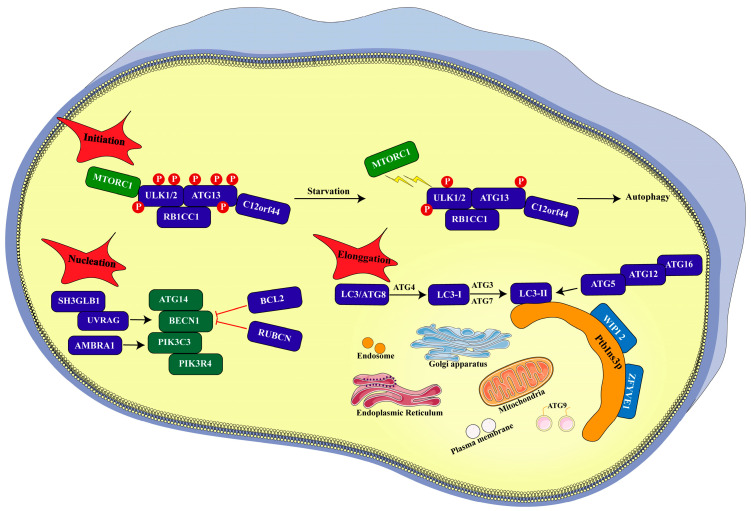
Diagrammatic representation of the autophagy process in mammals. **Initiation.** Nutrient deficiency leads to the dissociation of ULK1/2 complex and various ATG proteins from MTORC1, resulting in the dephosphorylation of the ULK1/2 complex and autophagy induction. **Nucleation.** ULK1 enhances the phosphatidylinositol-3-kinase activity of a multiprotein complex comprised of BECN1, PIK3C3/VPS34, PIK3R4/VPS15, ATG14, and NRBF2 and furtherly drives the autophagosome nucleation. Notably, SH3GLB1, UVRAG, and AMBRA facilitate the nucleation of autophagosomes, while BCL2 and RUBCN inhibit this process. **Elongation.** After WIPI2 binds to PtdIns3P complex and is activated, the ATG12-ATG5-ATG16L1 complex and LC3 are recruited, contributing to the expansion of phagophores.

**Figure 3 biomolecules-15-00915-f003:**
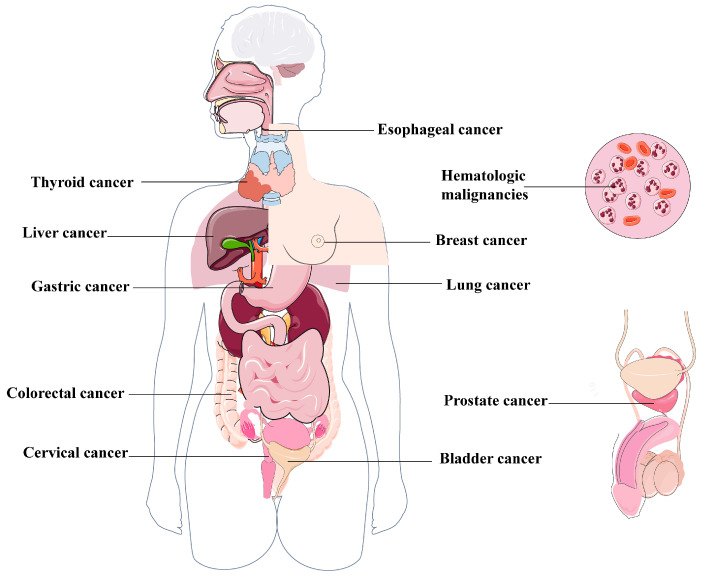
Several types of human cancers are linked to the dysregulation of autophagy. Autophagy is demonstrated to play a complex role in the occurrence and development of various human cancers, including lung cancer, breast cancer, esophageal cancer, colorectal cancer, prostate cancer, gastric cancer, liver cancer, thyroid cancer, cervical cancer, bladder cancer, hematologic malignancies, among others.

## Data Availability

Not applicable.

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
