# Peer review of "Autophagy: Shedding Light on the Mechanisms and Multifaceted Roles in Cancers"

_biomolecules, 2025, doi:10.3390/biom15070915_

Round 1

Reviewer 1 Report

Comments and Suggestions for Authors

This review primarily focuses on the role of autophagy in cancer progression. It discusses the classification of autophagy, the autophagic process, and how autophagy contributes to the induction or suppression of various cancer types through the expression or absence of autophagy-related markers. However, this review requires significant revisions and improvements before it can be considered for publication.

  1. Overall, the manuscript contains numerous grammatical errors, awkward phrasing, and unclear sentence constructions that affect the overall readability and scientific clarity. I recommend a thorough revision of the manuscript for English grammar, sentence structure, and style.
  2. Line 50: The word “essential” is used twice. Please revise the sentence to avoid repetition.
  3. Lines 56–75: This paragraph mainly discusses the global distribution of different cancer types without clearly connecting it to the role of autophagy. I suggest summarizing the cancer distribution briefly and focusing more on how autophagy contributes to the progression of each cancer type, if the intention is to explore cancer-specific roles.
  4. Lines 77–83: The discussion on the dual role of autophagy in cancer lacks appropriate references. Please include relevant citations to support these statements.
  5. Generally, the introduction lacks depth and clarity. It does not provide a comprehensive overview of autophagy or its roles in various diseases. Much of the content appears surface-level, with some unnecessary details. I recommend revising the section to offer a clearer, more structured introduction to autophagy and its relevance in disease contexts.
  6. Lines 89–91: Please add relevant citations.
  7. Please pay close attention when writing gene and protein names to avoid confusion and to convey the correct meaning. Carefully review all gene and protein nomenclature and follow standard conventions consistently. For example, gene names like BECN1 (human) and Becn1 (mouse) should be italicized, while protein names (e.g., Beclin 1) should not.
  8. The authors discuss how the expression of certain autophagic markers (ATG5, ATG7, Beclin1) may induce or suppress various cancer types. However, the underlying mechanisms are not well explained. I suggest providing more detail on how these markers mechanistically contribute to cancer progression or suppression.
  9. How does the induction or suppression of autophagy influence cancer survival? I recommend including relevant clinical data or studies that demonstrate how the presence or absence of key autophagy-related markers affects cancer progression. Incorporating such real-world evidence would strengthen the review.
  10. Lines 663–666: Please review this section, as the text appears unnecessary or out of context.
  11. Lines 683-691: The same information is repeated twice. “Hence, in the foreseeable future, it is plausible to explore the potential of integrating conventional cancer treatment with the modulation of autophagy activity. This can involve utilizing autophagy inducers or inhibitors tailored to specific tumor development and stages, thereby offering a promising avenue for anticancer therapy.” Please correct. Also please use proper, readable words and phrases for quick and clear delivery of ideas. Also, please use clear and precise language to ensure quick and effective communication of ideas. Avoid overly complex or redundant phrases.
  12. The Discussion and Future Directions section does not clearly address the gap between current applications of autophagy and its potential use in cancer therapy. Since the review focuses on autophagy as a therapeutic target in cancer, I suggest elaborating on this gap and discussing how future research could bridge it.
Comments on the Quality of English Language

This review contains numerous grammatical errors, awkward phrasing, and unclear sentence constructions that affect the overall readability and scientific clarity. I recommend a thorough revision of the manuscript for English grammar, sentence structure, and style.

Author Response

Comment1:Overall, the manuscript contains numerous grammatical errors, awkward phrasing, and unclear sentence constructions that affect the overall readability and scientific clarity. I recommend a thorough revision of the manuscript for English grammar, sentence structure, and style.

Response1:Thank you very much for reviewing our manuscript and providing valuable comments. To address these crucial language issues, we submitted our manuscript to a professional native English editing service for comprehensive and in-depth polishing. We believe that after this professional and thorough revision of the English language, the readability of the manuscript and the clarity of scientific expression will be significantly enhanced. The relevant changed content has been highlighted in yellow text.

Comment2:Line 50: The word “essential” is used twice. Please revise the sentence to avoid repetition.

Response2:Thanks for your careful review. In the revised manuscript, we have removed the repeated word "essential" and checked all the sentences in the text to prevent similar mistakes from occurring.

Comment3:Lines 56–75: This paragraph mainly discusses the global distribution of different cancer types without clearly connecting it to the role of autophagy. I suggest summarizing the cancer distribution briefly and focusing more on how autophagy contributes to the progression of each cancer type, if the intention is to explore cancer-specific roles.

Response3:Thank you for your constructive suggestion. Based on your suggestions, we have significantly reduced the description of cancer distribution and focused more on how autophagy functions at different types and stages of tumors in the revised manuscript. In addition, we also provide examples to illustrate that the role of autophagy in tumors is autophagy gene-specific and tumor-specific. The relevant changed content has been highlighted in yellow text.

Comment4:Lines 77–83: The discussion on the dual role of autophagy in cancer lacks appropriate references. Please include relevant citations to support these statements.

Response4:We are appreciated for your suggestions. In the revised manuscript, we have added appropriate references to support the dual role of autophagy in different cancer types and stages. The relevant changed content has been highlighted in yellow text.

Comment5:Generally, the introduction lacks depth and clarity. It does not provide a comprehensive overview of autophagy or its roles in various diseases. Much of the content appears surface-level, with some unnecessary details. I recommend revising the section to offer a clearer, more structured introduction to autophagy and its relevance in disease contexts.

Response5:Thank you very much for your in-depth and meticulous review of the manuscript, especially for the valuable suggestions you provided regarding the introduction section. We sincerely apologize for the shortcomings of the previous introduction in terms of depth, clarity, structure and focus of content. To meet your requirements and enhance the quality of the introduction, we have made substantial revisions to it in revised manuscript. Firstly, we deleted some unnecessary details, such as the extensive description of cancer distribution and focused on the interaction between autophagy and cancer. Secondly, we simply introduced the major types and process of autophagy. In addition, we elaborately introduced how autophagy plays different roles in various stages of tumor development and in different types of tumors. Finally, we added the content about the current clinical application and mechanism of action of autophagy modulators in tumors, as well as their future development directions. We believe that through the above substantive and comprehensive revisions, the revised introduction will significantly enhance the depth and clarity of the elaboration on the fundamentals of autophagy and its disease relevance and provide a more comprehensive and structured background. The relevant changed content has been highlighted in yellow text.

Comment6:Lines 89–91: Please add relevant citations.

Response6:Thank you for your reminder. According to the comments from other reviewers, we have polished the sentences in lines 89-91. Meanwhile, in accordance with your suggestions, we have added relevant citations in revised manuscript.

Comment7:Please pay close attention when writing gene and protein names to avoid confusion and to convey the correct meaning. Carefully review all gene and protein nomenclature and follow standard conventions consistently. For example, gene names like BECN1 (human) and Becn1 (mouse) should be italicized, while protein names (e.g., Beclin 1) should not.

Response7:Thanks for your careful review. In revised manuscript, we have carefully checked and revised the names of genes and proteins throughout the entire text to ensure that human gene names are written in capital letters and italics, mouse gene names in italics with the first letter capitalized, and protein names in non-italicized type. The relevant changed content has been highlighted in yellow text.

Comment8:The authors discuss how the expression of certain autophagic markers (ATG5, ATG7, Beclin1) may induce or suppress various cancer types. However, the underlying mechanisms are not well explained. I suggest providing more detail on how these markers mechanistically contribute to cancer progression or suppression.

Response8:We sincerely thank the reviewer for highlighting the lack of mechanistic details regarding the roles of ATG5, ATG7, and Beclin1 in cancer progression/suppression. In revised manuscript, we have supplemented the underlying mechanism descriptions of these autophagy markers in tumor progression or suppression. The relevant changed content has been highlighted in yellow text.

Comment9:How does the induction or suppression of autophagy influence cancer survival? I recommend including relevant clinical data or studies that demonstrate how the presence or absence of key autophagy-related markers affects cancer progression. Incorporating such real-world evidence would strengthen the review.

Response9:We sincerely appreciate your insightful suggestion. In the newly added section 5. The possibility of developing autophagy modulators for cancer treatment, we first expound the dual role of autophagy in the occurrence and development of tumors, and then introduce the current clinical application status of autophagy regulators and their impact on the survival period of patients. For example,a study reported that The survival period of 18 glioma patients who received combined treatment with chloroquine and anti-tumor drugs was significantly longer than that of the control group (33 months versus 11 months)[1]. Additionally, preoperative combined use of hydroxychloroquine and anti-tumor drugs can reduce CA199 levels in pancreatic cancer patients by 61%[2].There are also clinical reports showing that hydroxychloroquine enhances the anti-cancer activity of bevacizumab on glioblastoma by suppressing autophagy[3]. More clinical studies on how autophagy affects patient survival have been detailedly described in the revised section 5. The possibility of developing autophagy modulators for cancer treatment. Finally, we summarized the challenges that autophagy faces in the clinical treatment of cancer. The relevant changed content has been highlighted in yellow text.

Comment10:Lines 663–666: Please review this section, as the text appears unnecessary or out of context.

Response10:Thank you for your careful review. Lines 663-666 are indeed redundant. We are sorry for our mistake. We have removed this part in the revised manuscript. Thank you again for your reminder.

Comment11:Lines 683-691: The same information is repeated twice. “Hence, in the foreseeable future, it is plausible to explore the potential of integrating conventional cancer treatment with the modulation of autophagy activity. This can involve utilizing autophagy inducers or inhibitors tailored to specific tumor development and stages, thereby offering a promising avenue for anticancer therapy.” Please correct. Also please use proper, readable words and phrases for quick and clear delivery of ideas. Also, please use clear and precise language to ensure quick and effective communication of ideas. Avoid overly complex or redundant phrases.

Response11:Thank you for your reminder. In the revised manuscript, we have deleted the repetitive information. In addition, we have tried to remove the related complex phrases without changing the ideas we want to express. Furthermore, we have invited professional English editing service to polish our language of the entire text, making the expression more fluent and clearer. Thank you again for your suggestions.

Comment12:The Discussion and Future Directions section does not clearly address the gap between current applications of autophagy and its potential use in cancer therapy. Since the review focuses on autophagy as a therapeutic target in cancer, I suggest elaborating on this gap and discussing how future research could bridge it.

Response12:We deeply appreciate your critical insight regarding the insufficient discussion on translational gaps in autophagy-targeted cancer therapy. In the revised manuscript, we have substantially revised the discussion in Discussion and Future Directions section, providing a more in-depth analysis of the current applications and associated challenges of clinical autophagy modulators. Furthermore, we have explored potential strategies to overcome these challenges in future research and development. For example, many pharmacological autophagy regulators in applications face challenges related to poor target selectivity and insufficient specificity. These limitations can result in the induction of off-target effects and adverse reactions. Targeting specific complexes (such as ULK and VPS34), utilizing PROTACs/gene editing[4], and designing tumor-targeted delivery systems represent key approaches to advance the development of precise autophagy regulators. Another challenge is that commonly used biomarkers for monitoring autophagy are not always applicable for monitoring autophagic flux (the degradation products of autophagosomes and their contents in lysosomes). Dual fluorescence-labeled LC3 can be effectively utilized to monitor intracellular autophagic flux in real-time[5-7]. The revised Discussion and Future Directions section provides additional details on the specific content. The relevant changed content has been highlighted in yellow text. 

  1. Briceno E, Reyes S, Sotelo J. Therapy of glioblastoma multiforme improved by the antimutagenic chloroquine. Neurosurg Focus. 2003; 14: e3.
  2. Boone BA, Bahary N, Zureikat AH, Moser AJ, Normolle DP, Wu WC, et al. Safety and Biologic Response of Pre-operative Autophagy Inhibition in Combination with Gemcitabine in Patients with Pancreatic Adenocarcinoma. Ann Surg Oncol. 2015; 22: 4402-10.
  3. Liu LQ, Wang SB, Shao YF, Shi JN, Wang W, Chen WY, et al. Hydroxychloroquine potentiates the anti-cancer effect of bevacizumab on glioblastoma via the inhibition of autophagy. Biomed Pharmacother. 2019; 118: 109339.
  4. Li X, Song Y. Proteolysis-targeting chimera (PROTAC) for targeted protein degradation and cancer therapy. J Hematol Oncol. 2020; 13: 50.
  5. Galluzzi L, Bravo-San Pedro JM, Levine B, Green DR, Kroemer G. Pharmacological modulation of autophagy: therapeutic potential and persisting obstacles. Nat Rev Drug Discov. 2017; 16: 487-511.
  6. Klionsky DJ, Abdelmohsen K, Abe A, Abedin MJ, Abeliovich H, Acevedo Arozena A, et al. Guidelines for the use and interpretation of assays for monitoring autophagy (3rd edition). Autophagy. 2016; 12: 1-222.
  7. Kaizuka T, Morishita H, Hama Y, Tsukamoto S, Matsui T, Toyota Y, et al. An Autophagic Flux Probe that Releases an Internal Control. Mol Cell. 2016; 64: 835-49.

Reviewer 2 Report

Comments and Suggestions for Authors

The review presented by You et al, is describing autohagy dishomeostasis in several forms of cancers. While the theme is of general interest the review remained a bit superficial since contents are digested fast.

  • the authors focused in providing information of autophagy alteration only in solid tumors. It could be of interest providing a least one example of blood neoplasia in which autopaghy alteration can drive tumor inception, progression and survival
  • A paragraph discussion the possibility to develop novel autophagy modulators to treat cancers is not present and must be considered.
  • authors focused only on bulk autophagy. Selective autophagy are involved as well. For instance, Chipurupalli et al, demonstrated how in breast cancer ER-phagy contributes to cancer survival in hypoxic environment. Such information are important. the review cannot be limited to general autophagy.

Author Response

Comment1:the authors focused in providing information of autophagy alteration only in solid tumors. It could be of interest providing a least one example of blood neoplasia in which autopaghy alteration can drive tumor inception, progression and survival

Response1:Thank you for highlighting this gap. To further enrich the content of our manuscript, we have incorporated Section 4.6 Hematologic Malignancies in the revised manuscript, focusing on the dual role of autophagy in hematological malignancies, including but not limited to chronic myeloid leukemia, acute promyelocytic leukaemia, and multilple myeloma.The relative contents have been highlighted in yellow text.

Comment2:A paragraph discussion the possibility to develop novel autophagy modulators to treat cancers is not present and must be considered.

Response2:Thanks for your valuable suggestion. We have added a dedicated section 5. The possibility of developing autophagy modulators for cancer treatment on emerging autophagy modulators in revised manuscript. In this section, we discussed how the classical autophagy inhibitor 3MA and hydroxychloroquine exert their anti-cancer effects in various cancers by inhibiting autophagy. While the clinical application of traditional autophagy modulators is often restricted due to their non-specific accumulation and systemic toxicity. Thus, we also discussed the breakthrough directions for novel autophagy regulators. For example, the targeted protein degradation platform ATNC (phagosome) based on the autophagy pathway utilizes nanobodies to achieve selective target recognition[1]What’s more, Cheng et al. developed GPX4-AUTAC, which precisely degrades the GPX4 protein through the autophagy pathway. By mimicking the natural ubiquitination-autophagy receptor recognition mechanism (TRAF6/p62), it selectively induces autophagy-dependent degradation of GPX4, thereby efficiently triggering ferroptosis and inhibiting the growth of breast cancer[2]. In revised manuscript, we discussed the dual role of autophagy in cancer treatment, as well as how to utilize autophagy regulators to enhance the efficacy of existing therapies or overcome drug resistance. We also discussed the potential autophagy regulators in current research, including classical regulators and new ones, and explained their mechanism of action, preclinical research results, and potential clinical applications. We believe that through our revision of content, readers will gain a more complete understanding of the regulators of autophagy and its development directions. The relevant changed contents have been highlighted in yellow text. Thank you again sincerely for your suggestions on our manuscript.

Comment3:authors focused only on bulk autophagy. Selective autophagy are involved as well. For instance, Chipurupalli et al, demonstrated how in breast cancer ER-phagy contributes to cancer survival in hypoxic environment. Such information are important. the review cannot be limited to general autophagy.

Response3:We sincerely appreciate the comment on selective autophagy. In revised Section2. Classification of Autophagy, we introduced the related concepts and types of organelle autophagy, e.g. mitophagy, ER-phagy etc.. Meanwhile, in revised Section4. Links between autophagy and cancers, we have added description of the multiple roles of organelle autophagy in various cancers. For example, we added the content on how mitophagy increased the therapeutic sensitivity of lung cancer to cisplatin-induced apoptosis and the activation of mitophagy may lead to resistance of liver cancer cells to cabozantinib in liver cancer. In addition, we have incorporated the literature you recommended and further elaborated on the dual role of ER-phagy in breast cancer, thereby enhancing the comprehensiveness of the article. The relevant changed contents have been highlighted in yellow text.

  1. He H, Zhou C, Chen X. ATNC: Versatile Nanobody Chimeras for Autophagic Degradation of Intracellular Unligandable and Undruggable Proteins. J Am Chem Soc. 2023; 145: 24785-95.
  2. Gong R, Wan X, Jiang S, Guan Y, Li Y, Jiang T, et al. GPX4-AUTAC induces ferroptosis in breast cancer by promoting the selective autophagic degradation of GPX4 mediated by TRAF6-p62. Cell Death Differ. 2025.

Reviewer 3 Report

Comments and Suggestions for Authors

The role of autophagy is a current subject of research either in physiology or in therapeutics, thus this text is valuable for research on several fields, among them in Cancer; however, some points should be clarified:

  1. The line “It is worth noting…” on L20-21 is an odd statement which should be deleted as it is not sustained by experimental evidence.
  2. Also, the statement “…one kind of programmed cell death” on line 38 was not proposed in the respective research and should also be eliminated.
  3. The so called “classification of autophagy” is only a proposal of this text, not an accepted classification (P3, L93 and P2, L89).
  4. The statement of “inhibition” and “promotion” of malignant cell growth is a dual and opposite referral which bring confusion, most current therapeutic studies in cancer disclose the idea of autophagia inhibition as a potential treatment, I do not know of any study which states that autophagia stimulation can be a sustained therapy for cancer. The authors must clarify or discuss this controversial point raised in the current text. (e.g. P15,L681; P16, L685).

Author Response

Comment1:The line “It is worth noting…” on L20-21 is an odd statement which should be deleted as it is not sustained by experimental evidence.

Response1:Thanks for your advice. According to your suggestion, we have deleted the relevant content in revised manuscript.

Comment2:Also, the statement “…one kind of programmed cell death” on line 38 was not proposed in the respective research and should also be eliminated.

Response2:Thank you for your rigorous suggestions and we apologize for citing this inaccurate term. We have deleted the description in revised manuscript.

Comment3:The so called “classification of autophagy” is only a proposal of this text, not an accepted classification (P3, L93 and P2, L89).

Response3:We would like to express our appreciation to you for precise suggestion on how to improve this paper. Based on your suggestion, we revised "Classification of Autophagy" to "Major Types of Autophagy" in the revised manuscript and clearly describes the criteria for our current classification and refined the sentence structure, thereby significantly improving the precision and scientific rigor of the presentation. Once again, we would like to express our sincere gratitude for your advice.

Comment4:The statement of “inhibition” and “promotion” of malignant cell growth is a dual and opposite referral which bring confusion, most current therapeutic studies in cancer disclose the idea of autophagia inhibition as a potential treatment, I do not know of any study which states that autophagia stimulation can be a sustained therapy for cancer. The authors must clarify or discuss this controversial point raised in the current text. (e.g. P15,L681; P16, L685).

Response4:We sincerely thank you for raising this crucial and insightful point and we apologize for any confusion caused by our ambiguous language expression. In the revised Section 6. Discussion and future directions, we further polished the manuscript's language expression, explicitly emphasizing that inhibiting autophagy represents the primary focus of current research and clinical development. Additionally, In the revised Section5. The possibility of developing autophagy modulators for cancer treatment and Section 6. Discussion and future directions, we strictly confined the expression of "autophagy inducers" to specific contexts, such as the prevention of early tumor initiation or specific auxiliary function. Eventually, we pointed that Future research should prioritize the development of more specific and effective autophagy regulators based on type, stage or genetic context of the cancers while conducting comprehensive investigations into the molecular mechanisms underlying autophagy regulation, thereby providing stronger guidance for clinical applications. The relevant changed content has been highlighted in yellow text.

Round 2

Reviewer 1 Report

Comments and Suggestions for Authors

The authors have successfully addressed all review comments and have significantly improved the quality of the manuscript through the addition of insightful content, improved organization, and clearer language. I believe the current version is suitable for publication.

However, I noticed that numbers appear after keywords. please remove them if they are not relevant.

Reviewer 2 Report

Comments and Suggestions for Authors

Upon revision, the manuscript has been substantially improved and it merits publication in this journal. 

Reviewer 3 Report

Comments and Suggestions for Authors

The manuscript has been considerably improved.